

# Evaluation and Comparison among Multiple Forcing Data Sets for Precipitation and Shortwave Radiation over Mainland China

Fan Yang[1], Hui Lu[1,4], Kun Yang[1,2,3,4], Wei Wang[1], Chengwei Li[1], Menglei Han[1], Yishan Li[1]

[1]Ministry of Education Key Laboratory for Earth System Modeling, Department of Earth System Science, Tsinghua University, Beijing, 100084, China
[2]CAS Center for Excellence in Tibetan Plateau Earth System, Beijing, 100101, China
[3]Key Laboratory of Tibetan Environment Changes and Land Surface Processes, Institute of Tibetan Plateau Research, Chinese Academy of Sciences, Beijing 100101, China
[4]The Joint Center for Global Change Studies, Beijing, 100875, China

*Correspondence to*: Hui Lu (luhui@tsinghua.edu.cn)

**Abstract.** Precipitation and shortwave radiation play important roles in climatic, hydrological and biogeochemical cycles. Currently, several global and regional forcing data sets, such as Global Land Data Assimilation System (GLDAS), China Meteorological Administration of land data assimilation system (CLDAS) and China Meteorological Forcing Dataset (CMFD), can provide estimates of these two variables for China. In this study, precipitation and shortwave radiation from CMFD, CLDAS and GLDAS were inter-compared with one another and against ground station observations during 2008-2014. The results demonstrate that all the three forcing data sets can capture the spatial distribution characteristics of precipitation over mainland China while CMFD is closer to the observed ground data. The temporal variations of precipitation from the three forcing data sets also match in most areas of mainland China with the observation, while CMFD and GLDAS perform better than CLDAS. Compared with the precipitation observation, CMFD has a slight overestimation but its correlation coefficient reaches 0.97. CLDAS precipitation shows a significant underestimation, with a bias of 107mm yr-1 in annual mean, while GLDAS has the closest annual mean precipitation to the observed data. As for shortwave radiation, CLDAS and GLDAS heavily overestimate shortwave radiation when compared against station data, while CMFD is consistent with observation data. Spatially, the three forcing data sets have some common distribution features. Compared with CMFD, CLDAS and GLDAS have higher radiation values in most areas of mainland China and the temporal variations difference of shortwave radiation from the three forcing data sets mainly exists the south of 34°N. In summary, precipitation estimates of CMFD and GLDAS are more credible and CMFD outperforms CLDAS and GLDAS in shortwave radiation estimation over mainland China. Findings from this study can provide guidance to communities regarding the performance of different forcing data over mainland China.

## 1 Introduction

Precipitation and shortwave radiation are the fundamental water and energy sources of land surface biological, physical and chemical processes (Zhao and Zhu 2015; Zhang et al. 2010). They can affect the moisture and heat exchange between the atmosphere and the land surface (Pan et al. 2014; Tian et al. 2007; Fekete et al. 2004; Gottschalck et al. 2005). Also, these



two variables are the basic meteorological forcing inputs for land process simulations such as crop simulation, hydrologic modeling, dryland expansion estimation and dust events analysis (Bart and Lettenmaier 2004; Tang et al. 2007, 2008; Huang et al. 2015; Kang et al. 2016). Therefore, accurate precipitation and shortwave radiation data are essential for studies of climate change and land surface processes.

Although conventional measurement at stations can obtain the "true value" of the measured variable, it can only represent local scale information (Maurer 2002, Bogh et al. 2003), and is unable to depict the characteristics of spatial variation completely and continuously due to the limited number and location of stations (Duan et al 2012). In the late 1980s, data assimilation technology was proposed to reconstruct high resolution forcing data of historical climate (Xie et al. 2011, Zhao et al. 2010), and this brought unprecedented opportunities for researchers. These forcing data sets, which usually include

precipitation, shortwave radiation, temperature, specific humidity, wind speed, surface pressure, and other meteorological data, are derived by assimilating numerical weather forecast information, ground observation data and remote sensing data together (Xie et al. 2011, Zhao et al. 2010, Pan et al. 2010). There are many forcing data sets currently available, such as that from the National Centers for Environmental Prediction and the National Center for Atmospheric Research reanalysis (NCEP/NCAR, Kalnay et al., 1996; Kistler et al., 2001), Global Land Data Assimilation System (GLDAS, Rodell et al.

2004), European Center for Medium-range Weather Forecasts (ECMWF) reanalysis ERA-Interim (Dee et al. 2011), Japanese 55-year Reanalysis (JRA-55, Kobayashi et al. 2015). In recent years, Chinese researchers have made great progress in developing forcing data. Two forcing data sets, namely China Meteorological Forcing Dataset (CMFD, released by the Institute of Tibetan Plateau Research, Chinese Academy of Sciences) and the China Meteorological Administration of land data assimilation system (CLDAS), which cover China have been produced. These forcing data sets are widely used because

of their high spatial resolution, long time span over large areas and convenience of obtaining and processing. For example, a CMFD forcing data set has been used to simulate the permafrost and seasonally frozen ground conditions on the Tibetan Plateau (Gao and Wang, 2013), to analyze precipitation impacts on vegetation spring phenology (Shen et al. 2015), to model land surface water and energy cycles of a mesoscale watershed (Xue et al. 2013), and to assess the climate and human impacts on surface water resources in the middle reaches of the Yellow River (Hu et al. 2015). Additionally, CMFD and

GLDAS forcing data sets have been used to improve land surface temperature modeling for dry land in China (Chen et al. 2011). GLDAS has also been applied to analyze the long-term terrestrial water storage variations in the Yangtze River basin (Huang et al. 2013), and a newly released CLDAS forcing data set has been adopted in a recent study of drought monitoring (Han, 2015).

However, as these products have merged multiple data sets, there are various errors caused by the numerical model, data

fusion scheme and observation system change in the forcing data sets (Xie et al. 2011). The bias associated with a forcing data set can be propagated into model results (Wang et al, 2016), which in turn may show unrealistic results if the forcing data sets are not reliable (Cosgrove et al., 2003). For example, errors in precipitation and shortwave radiation have a great impact on simulations of soil moisture, runoff and heat fluxes (Luo et al., 2003). As a result, it is necessary to evaluate the accuracy of the data sets so that the bias of these forcing data sets are fully recognized before they can be applied to land



surface studies (Pan et al. 2014).Some studies have been conducted to evaluate the forcing data sets. Wang et al. (2014, 2016) assessed the applicability of GLDAS monthly precipitation data in China from 1979 to 2012, and found that both GLDAS-1 and GLDAS-2 precipitation matched well with observation precipitation data by visual inspection. Wang et al. (2011) validated the GLDAS-1/NOAH daily and monthly precipitation data in a mesoscale basin in northeastern China and

concluded that GLDAS was of high quality for daily and monthly precipitation during March 2003 to March 2006. Wang and Zeng (2012) evaluated six reanalysis products (i.e., MERRA, NCEP–NCAR, CFSR, ERA-40,ERA-Interim, and GLDAS-1) using in situ measurements at 63 weather stations over the Tibetan Plateau and the result showed that GLDAS had the best overall performance for both daily and monthly precipitation.

Though the quality of GLDAS has been relatively well evaluated in previous studies, it is not bias free and the credibility of

GLDAS in recent years over continental China is still unclear. On the other hand, both CMFD and CLDAS are developed and maintained by Chinese scientists and they are supposed to have high accuracy and reliability because more ground observation data have been merged into them. However, so far, there is no comprehensive evaluation around CMFD and CLDAS forcing data sets. In this study, the performance of CMFD, CLDAS and GLDAS in terms of precipitation and shortwave radiation were inter-compared and evaluated against available in situ observation. Such inter-comparison will

benefit researchers to select meteorological forcing data, and in turn help the producer to further improve the quality of forcing data.

## 2 Used data

### 2.1 Forcing data sets

### 2.1.1 CMFD

The CMFD forcing data set was developed by the Institute of Tibetan Plateau Research, Chinese Academy of Sciences (**He and Yang, 2011**). The product covers the region of 70.05°E -140.05°E and 15.05°N -55.05°N, and includes precipitation, shortwave radiation, longwave radiation, temperature, specific humidity, wind speed and surface pressure. The Global Energy and Water cycle Experiment - Surface Radiation Budget (GEWEX-SRB) radiation data, Tropical Rainfall Measuring Mission (TRMM) 3B42 precipitation data, and GLDAS data provide the background of the reanalysis for the CMFD forcing

data. Gauge observation data from 740 stations of China Meteorological Administration (CMA) and shortwave radiation data estimated by the Hybrid model which was developed by **Yang et al. (2001, 2006)** are used to correct systematic deviations in reanalysis data. Other basic information of the data is listed in Table 1.

### 2.1.2 CLDAS-V2.0

CLDAS-V2.0 was developed by CMA (**Shi 2013**) and its spatial coverage is 70°E -150°E, 0°N -60°N. This is hourly gridded

data with a spatial resolution of 0.0625°×0.0625°. CLDAS includes land surface forcing data, such as precipitation,





shortwave radiation, temperature, specific humidity, wind speed and surface pressure, as well as soil status variables. It is a relatively new product, with current temporal coverage from 2008 to 2016. Shortwave radiation is retrieved from the FY-2C/E series of geostationary meteorological satellites. The Discrete Ordinates Radiative Transfer Program for a Multi-Layered Plane-Parallel Medium (DISORT) method is used in the retrievals for radiation transfer calculations. Precipitation is

combined and interpolated from two products, one is the Climate Prediction Center Morphing Technique (CMORPH) product and the other is an hourly integration product (V1.0) made by CMA which combines Chinese rain gauges and CMORPH products **(Zhu, 2016)**.

### 2.1.3 GLDAS-1

The 0.25°×0.25° monthly GLDAS-1 forcing data from the NOAH model is provided by the US National Aeronautics and

Space Administration (NASA). From 2001 to the present, this version was forced with a combination of National Oceanic and Atmospheric Administration (NOAA)/ Global Data Assimilation System (GDAS) atmospheric analysis fields, spatially and temporally disaggregated NOAA Climate Prediction Center Merged Analysis of Precipitation (CMAP) fields, and observation-based downward shortwave fields from the Air Force Weather Agency (AFWA) (Rui, 2015).

### 2.2 Observation data

### 2.2.1 Gridded precipitation data

In order to validate the precipitation products of the above-mentioned three forcing data sets, a 0.25°×0.25° gridded monthly precipitation data set (CN05.1) over mainland China is used as a reference (Wu and Gao 2013). This product was interpolated from more than 2000 gauge stations over mainland China, and an "anomaly approach" (New et al. 2000) was applied in the interpolation. This can provide a relatively precise estimation of precipitation with high spatial resolution as a

large amount of observation data was used.

### 2.2.2 Shortwave radiation station data

Because there is no available gridded shortwave radiation observation data, we used two kinds of station data to verify the forcing data sets. The first kind is from CMA, but attention must be paid to CMFD, as their radiation is estimated from a hybrid model and thus is not fully independent of the radiation data used for evaluation. The other kind is from independent

station data which are not included in the forcing data sets.

1) Shortwave radiation station data from CMA

A daily surface solar radiation data set from a total of 716 CMA stations updated to 2010 is offered by the Data Assimilation and Modeling Center for Tibetan Multi-spheres (http://dam.CMFD.ac.cn/). This data set merges the hybrid model estimate at 716 CMA stations with the ANN-based model estimate at 96 radiation stations which, because of its high accuracy, was used



to correct the hybrid model estimate dynamically at a monthly scale (Tang et al. 2013). In this paper, we selected 625 stations with full data for 2008-2010.

2) Independent station data

① Shortwave radiation station from CERN

The Chinese Ecosystem Research Network (CERN) was established in 1988 by the Chinese Academy of Sciences (Su et al. 2005). The 2008-2014 shortwave radiation observation data used in this paper are provided by 35 field experimental stations in CERN covering various ecosystems, including farmland, forest, grassland, lakes and the sea. As shown in Fig. 1, these stations are located evenly over mainland China and cover various climate types and land cover types. Meanwhile, CERN stations are independent of CMA stations which are partly used in CMFD and CLDAS. Therefore, CERN is a perfect
reference to assess the performance of these three forcing data sets, although the gauge density is not so high.

② Shortwave radiation station from HiWATER

Shortwave radiation observation data from eight stations in the Heihe River have beeen collected from the Heihe Watershed Allied Telemetry Experimental Research (HiWATER, Xin et al. 2013) and it is widely used for land surface process studies (Liu et al. 2016; Cheng et al. 2014). There are 2-3 sites distributed in the upper, middle and downstream of the Heihe River
basin.

③Shortwave radiation station from the TPE Database

A daily shortwave radiation record from the Meteorological dataset of the Ngari Desert Observation and Research Station and from the Meteorological dataset of the Muztagh Ata Station for Westerly Environment Observation and Research, was obtained from the Third Pole Environment (TPE) Database (http://www.tpedatabase.cn), as shown in Figure 1.These two
stations were used as a supplement to evaluate the performance of the three reanalysis data sets on the west Tibetan Plateau where there are very few CMA ground stations.

## 3 Methodology

Precipitation and shortwave radiation were evaluated from various spatial and temporal scales. In terms of spatial scale, the patterns of each data were compared and three metrics were computed, i.e. average precipitation over mainland China
(Mean), standard deviation (SD) and coefficient of variation (CV), the latter two can reflect the degree of dispersion of the data set itself. In order to make a further comparison, CMFD and CLDAS were resampled to 0.25°×0.25° which is the same as the CN05.1 by the Nearest Neighbor (Hu et al. 2013) resampling method. Average bias over mainland China, correlation coefficient, root mean square error (RMSE) and relative bias between forcing data set and CN05.1 were also calculated. The correlation coefficient evaluates the correlation between two data sets. RMSE is a frequently used measurement of
differences between two data sets. Relative bias reflects the degree to which the measured value is over- or under-estimated. The formulations of these metrics are as follows:




$$SD = \sqrt{\frac{1}{N}\sum_{i=1}^{N}(x_i - \bar{x})^2} \tag{1}$$

$$CV = \frac{SD}{\bar{x}} \tag{2}$$

$$\text{Correlation coefficient} = \frac{\sum_{i=1}^{N}(x_i - \bar{x})(y_i - \bar{y})}{\sqrt{\sum_{i=1}^{N}(x_i - \bar{x})^2 \sum_{i=1}^{N}(y_i - \bar{y})^2}} \tag{3}$$

$$RMSE = \sqrt{\frac{\sum_{i=1}^{N}(x_i - y_i)^2}{N}} \tag{4}$$

$$\text{Relative bias} = \frac{\sum_{i=1}^{N} x_i}{\sum_{i=1}^{N} y_i} - 1 \tag{5}$$

$$TSD = \sqrt{\frac{1}{m}\sum_{j=1}^{m}(t_j - \bar{t})^2} \tag{6}$$

$$TCV = \frac{TSD}{\bar{t}} \tag{7}$$

where $x_i$ is the element of data sets and $\bar{x}$ is the average value of this data set; $y_i$ is the element of the reference data set and $\bar{y}$ is the average for data $y_i$; $N$ is the number of points in the data; $m$ is the number of months during 2008 to 2014; $t_j$ is precipitation or shortwave radiation value of per month of each grid point in the data sets and $\bar{t}$ is the average of $t_j$.

For temporal scale, monthly anomalies of precipitation were derived by subtracting the seven-year monthly climatology from each data element. Temporal standard deviation (TSD) and temporal coefficient variation (TCV) were computed to demonstrate the fluctuation characteristics of the forcing data sets in a time series. The larger these values, the greater the temporal difference.

To verify the accuracy of shortwave radiation for each of the forcing data sets using gauge observations data, a pixel-point method (Chen et al. 2013) was applied by pairing gauge observation data from the corresponding grids of these three forcing products. Correlation coefficient, RMSE and relative bias were selected as evaluation metrics of the pixel-point comparison. Then we used the same method as that used for precipitation to compare shortwave radiation of the three forcing data sets.

## 4 Evaluation of precipitation data

### 4.1 Spatial distribution of precipitation

It is obvious that all the data sets reveal a gradual increasing pattern of annual mean precipitation from northwest to southeast in mainland China (Fig. 2). The distribution of precipitation from CMFD is similar to that of CN05.1, which is used as a reference, even in the details, such as the depiction of transition zone on the Tibetan Plateau. Compared with CN05.1, the 600mm yr-1 rainfall contour of CLDAS is more southward, and the area where the annual mean precipitation is higher than 1500mm is smaller. GLDAS shows that precipitation in several regions of Fujian province and Zhejiang



province is higher than 2400 mm yr-1, which is not found in other products. Besides, the distribution of CLDAS and GLDAS in Tibet are quite different from the reference.

As shown in Table 2, CLDAS and GLDAS show an underestimation from compared to the observation data, especially for the Mean value of CLDAS which is 103mm yr-1 lower than CN05.1. On the contrary, Mean precipitation is slightly

overestimated by CMFD. As for SD and CV, CMFD and GLDAS are closer to CN05.1.

The comparison results with statistical metrics and the frequency distribution of the bias are listed in Fig. 3. The most remarkable difference between the forcing data sets and CN05.1 is mainly located in the south of 34 ° N, where precipitation was relatively abundant. This pattern indicates that the quality of the forcing data sets in north China is better than in the south. From the histogram of bias (Fig. 3 (d)), it is clear that for CMFD most regions have a relatively small positive bias

which means CMFD slightly overestimates annual precipitation in some regions. This result is consistent with that of Wang et al. (2016), who calculated the correction of CMFD daily precipitation data over the Qinghai-Tibetan Plateau from 2009 to 2012 and found that it mainly showed an overestimation for more than 255 days of the year. The area where the bias is greater than 200mm yr-1 in CMFD accounts for less than 2% of the total area of mainland China. Moreover, for CLDAS and GLDAS, the areas with a negative bias is much larger. CLDAS underestimated precipitation in about 80% of the areas over

mainland China, especially in southeastern China. However in large areas of the Tibetan Plateau CLDAS shows a heavy overestimate. Basically, GLDAS shows a similar pattern with CLDAS in western China, but there were some "hot spots" with significantly higher or lower value than CN05.1 interspersed in the southeast region. Notably, the correlation coefficient between CMFD and CN05.1 shown in Table 2 is the highest, reaching 0.97, while the absolute values of RMSE and relative bias are the lowest. For average bias, the absolute value of GLDAS is the smallest, while that of CLDAS is much greater

than the other two. As indicated above, better agreements with CN05.1 are observed for CMFD. From this perspective, CMFD outperforms CLDAS and GLDAS.

### 4.2 Temporal variation of monthly precipitation data

Figure 4 shows the time series of monthly mean precipitation anomalies averaged over the mainland China. The forcing data sets match well with CN05.1, as indicated by the similar inter-annual and decadal variability. However, for the difference

between the forcing data sets and CN05.1, during all 84 months CMFD has the smallest difference for 26 months, CLDAS has 18 months and GLDAS has 40 months, while CMFD has the largest difference for 26 months, CLDAS has 42 months and GLDAS has 16 months. In addition, the RMSE, relative bias and correlation coefficient between precipitation anomalies of each forcing data set and CN05.1 are shown in Table 3. CMFD has the smallest relative bias while GLDAS has the smallest RMSE and highest correlation coefficient. This result implies that the anomaly of GLDAS is best matched with

observation data which means GLDAS captures the seasonality better than the other two products, while CLDAS is the worst. In addition, a relatively high positive anomaly in August and September 2014 shown in CMFD.

The distribution characteristic of TSD is similar to the spatial distribution of annual mean precipitation, i.e. increasing from northwest to southeast. This pattern is reasonable as standard deviation usually increases with mean value. It can be seen



from Fig. 5 that the TSD of the four data sets are similar to each other, especially in the areas where TSD is higher than 105mm yr-1. However, there are obvious differences over the Tibetan Plateau, where the TSD varies greatly in this area. Additionally, CMFD is the closest to the observation data and the Mean value of CLDAS is the smallest, which relates to its lower annual mean precipitation.

The TSD of each grid is divided by the average precipitation during the seven-year period to obtain the TCV as shown in Fig. 6. It can be seen that the spatial distribution of TCV of the four data sets are similar to each other in southeast China. Southeast and northwest mainland China generally have smaller TCV values, while in other regions TCV is relatively larger. The TCV of CMFD is similar to that of CN05.1 and CMFD has a precise estimation on Mean value of TCV, with a slight difference from observations. However, the TCV of CLDAS and GLDAS are higher than that of CN05.1 in most areas of

north China. Moreover, their TCV is higher than 1.5 in dry regions (Xinjiang, Gansu and Inner Mongolia), while the observed TCV is not as high. Thus it is clear from the analysis that the TCV estimated by CMFD is closest to the observation values among the three products.

## 5 Evaluation of shortwave radiation data

### 5.1 Comparison against independent ground measurements

A comparison between gauge observations and the three products was carried out to examine which one was closest to the ground observations. As shown in Fig. 7 (a),(d) and (g), compared with the stations from CMA, the points of CMFD are distributed around the diagonal evenly, while CLDAS and GLDAS have a much higher shortwave radiation value in about 96% points, and their RMSE is almost 6 times that of CMFD. Though all the forcing data sets have a positive relative bias, meaning that they all overestimate the shortwave radiation value, the extent of overestimation by CMFD is slight.

Additionally, the correlation coefficient of the forcing data sets and observation all exceed 0.8, especially for CMFD which almost reaches 1. The high correlation may be due to the use of these site observations when producing the forcing data sets, leading the comparison results do not convince enough.

To make up for this problem, we use the observation data from CERN, which is not merged into these forcing data sets, and what stands out in Fig. 7 (b), (e) and (h) is that the shortwave radiation of most stations are significantly overestimated by

CLDAS and GLDAS, their RMSE values are about 4 times those of CMFD and even more than 10 times in terms of relative bias. On the contrary, CMFD coincides well with the observation data, the correlation coefficient value is the highest with a value of 0.93, and RMSE and relative bias are the smallest, which indicates that the estimation of CMFD for shortwave radiation is more precise than for CLDAS and GLDAS in these areas.

Due to the fact that the distribution of CERN observation stations relatively sparse in western China, this paper supplements

the data of eight observation stations in the Heihe River basin and two observation stations in the Tibetan Plateau to validate the three forcing data sets. As shown in Fig. 7 (c), (f) and (i), in these ten stations, both CLDAS and GLDAS show an obvious overestimation of shortwave radiation against gauge observation, while CMFD is closer to the in situ observation.



As for statistical indicators, the RMSE and relative bias of CLDAS and GLDAS are about 2 and 10 times that of CMFD, respectively. However, the correlation coefficient of GLDAS and observation is the highest followed by CMFD and CLDAS. In the study of Qi et al. (2015), GLDAS was also found to overestimates shortwave radiation from March 2000 to December 2007 in the Biliu Basin which is located in a coastal region of China. Besides, Wang et al. (2011) proved that shortwave radiation at the Changchun, Shenyang and Yanji stations in China was also overestimates by GLDAS from 2000 to 2006.

## 5.2 Spatial distribution of shortwave radiation

As mentioned above, the shortwave radiation data of CMFD matches well with the station data and so we used this as reference data to evaluate the performance of CLDAS and GLDAS over mainland China.

As can be seen from Fig. 8, the distribution of shortwave radiation of the three forcing data sets have some common characteristics. They all have a similar spatial pattern showing that the shortwave radiation in western China is higher than in the east, while the largest shortwave radiation value appears in the Tibetan Plateau and the value in northeast China is relatively low. It is clear from Fig. 8 (a) that for CMFD's estimation, two threshold values 170 W m-2 and 200 W m-2 divide the entire mainland China into three parts. In contrast, CLDAS is similar to GLDAS, the 200 W m-2 threshold line is offset to the northeast, and only a small part of Heilongjiang province and Chongqing province is less than 170 W m-2. In addition, the Mean values of CLDAS and GLDAS are higher than CMFD, while its degree of dispersion was relatively larger as indicated by a smaller SD and CV (Table 4).

The difference among the three forcing data sets is shown in Fig. 9. It can be found that CLDAS and GLDAS have a much higher shortwave radiation than CMFD in most regions. More than 95% of the area over mainland China shows a positive difference for CLDAS when compared with CMFD, especially in some areas of Xinjiang province and the area 24°N-44°N, 105°E-120°E. When compared with CMFD, GLDAS is also significantly higher except over the Tibetan Plateau and a very small area of Inner Mongolia. In terms of the statistical metrics shown in Table 4, the absolute value of average difference, RMSE, and the relative bias between CMFD and CLDAS are smaller but there is no significant difference with the metrics between CMFD and GLDAS, and the correlation coefficient of the three forcing data sets are all around 0.9. Overall, CLDAS and GLDAS are similar not only in spatial distribution but also in values, both being higher than CMFD in most regions of mainland China.

## 5.3 Temporal variation of shortwave radiation

As shown in Fig. 10, the change trend of anomaly of CMFD and CLDAS match well with each other. But the anomaly amplitudes of these two data sets are different. The anomaly of CLDAS is always positive after year 2013, which indicates that the shortwave radiation estimated by this product is clearly higher than the climatology mean value. This phenomenon also appears in GLDAS in most of the months of 2010-2014. Also, the magnitude of the fluctuation of GLDAS is larger than others. There are 52 months when the difference between GLDAS and CMFD is greater than that of CLDAS. As for the statistical metrics shown in Table 5, both the relative bias values are positive and the correlation coefficient are not high.



In Fig. 11 there is a clear pattern showing the TSD of the forcing data sets being divided into four grades by threshold 35 W m-2, 60 W m-2 and 80 W m-2. The TSD of northwest China is the largest and it gradually decreases from northwest to southeast. All forcing data sets show that the variations of shortwave radiation in the north is greater than that in the south. The difference among the three data sets mainly exists south of 34°N. GLDAS and CMFD are similar in Linzhi, the

5 southeast Tibetan Plateau, where both of their TSD are less than 35 W m-2. In particular, the TSD for CLDAS is smaller in south of 26°N, and its Mean value over mainland China is the smallest.

The spatial pattern of TCV for shortwave radiation of the three forcing data sets are similar, and all of them can be divided into four levels by value 0.15, 0.25 and 0.35, as shown by Fig. 12. The highest TCV appears in northwest and northeast China while the smallest TCV can be found in southwest and southeast China. To the south of 34°N, the TCV of CLDAS is

10 lower than 0.3 while CMFD is higher than 0.3 in the southeast. In addition, the estimation of TCV in CMFD and GLDAS has an obviously higher value in the vicinity of Sichuan province and Chongqing province than the surrounding areas. Similar to the distribution of TSD, the TCV of the three data sets are similar to each other north of 34°N and the difference mainly lies in the south of China. The common characteristics mentioned above are shown in Fig. 12.

## 6 Conclusions

In this study, precipitation and shortwave radiation data provided by CMFD, CLDAS and GLDAS were inter-compared and evaluated over mainland China. For precipitation, all the three forcing data sets reflect similar spatial distribution characteristics, i.e. a gradual increase from northwest to southeast in mainland China. The results also indicate that the spatial pattern of CMFD is closer to observation data than the other two data sets. When considering temporal variability, the monthly mean precipitation anomalies of the three forcing data sets match well with observation data and GLDAS is most

similar to CN05.1. The TSD and TCV are closely related to the amount of precipitation, and their distribution characteristics and values also show that CMFD and GLDAS perform better than CLDAS. Compared with the observation data, CMFD shows a slight overestimation but its RMSE and correlation coefficient perform best. CLDAS shows a significant underestimation over mainland China although it has a relatively high correlation with CN05.1. It is also worth noting that GLDAS has the closest annual mean precipitation to the observed data and differs by about only 3mm.

In terms of shortwave radiation, comparisons against ground shortwave radiation observation show that the shortwave radiation value is significantly overestimated by CLDAS and GLDAS, usually they have a much higher RMSE and relative bias value but relatively low correlation coefficient. However, CMFD is closer to the observation data and most of its statistical metrics perform much better. All forcing data sets show higher values in western China than in eastern China, and the biggest shortwave radiation value exists in the Tibetan Plateau. Also, the spatial characteristics of CLDAS and GLDAS

are similar, with both of them being higher than CMFD in most areas of mainland China. There is an anomaly whereby GLDAS fluctuates heavily while CMFD is more stable. The temporal variability of the three forcing data sets is more similar




north of 34°N, while the difference in the south is larger. All the results reflect the fact that temporal variation in the north of China is larger than that in the south.

To sum up, for precipitation of CMFD and GLDAS are better over mainland China than CLDAS and for shortwave radiation of CMFD is best while CLDAS and GLDAS have a serious overestimation. As these products are widely used and being developed, our results could benefit researchers for forcing data selection and uncertainty quantification and also could provide clues for data producers to further improve their data sets. Meanwhile, the results of this inter-comparison highlights that big uncertainties exist in the currently available forcing data, especially in the west region of mainland China where the density of ground stations is low, and where there is a need to improve the quality of forcing data in these regions.

### Acknowledgements

This work was jointly supported by the National Basic Research Program of China (No.2015CB953703), the National Key Research and Development Program of China (2016YFA0601603), and the National Natural Science Foundation of China ( 41371328 & 91537210).

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





**Table 1. Basic information of data sets used in this paper.**

| Name | Type | Analyzed period | Available period | Variables | Spatial resolution | Number of sites |
|------|------|------|------|------|------|------|
| CMFD | forcing data set | 2008-2014 | 1979-2016 | precipitation; shortwave radiation | 0.1° | |
| CLDAS | forcing data set | 2008-2014 | 2008-2016 | precipitation; shortwave radiation | 0.0625° | |
| GLDAS | forcing data set | 2008-2014 | 2000-2016 | precipitation; shortwave radiation | 0.25° | |
| CN05.1 | observation data | 2008-2014 | 1961-2014 | precipitation | 0.25° | |
| CMA | observation data | 2008-2010 | Different at each site | shortwave radiation | | 625 |
| CERN | observation data | 2008-2014 | Different at each site | shortwave radiation | | 35 |
| HiWATER | observation data | Different at each site | Different at each site | shortwave radiation | | 8 |
| TPE Database | observation data | Different at each site | Different at each site | shortwave radiation | | 2 |





**Table 2.  Spatial statistical metrics of annual mean precipitation and precipitation bias from 2008 to 2014.**

| Metrics | CN05.1 | CMFD | CLDAS | GLDAS |
|---|---|---|---|---|
| Mean(mm yr$^{-1}$) | 612.09 | 637.65 | 508.58 | 609.44 |
| SD(mm yr$^{-1}$) | 497.61 | 511.09 | 429.50 | 506.55 |
| CV | 0.81 | 0.80 | 0.84 | 0.83 |
| Average bias (mm yr$^{-1}$) | -- | 16.89 | -98.24 | -1.35 |
| RMSE(mm yr$^{-1}$) | -- | 125.61 | 184.27 | 249.83 |
| Relative bias | -- | 0.11 | -0.08 | 0.12 |
| Correlation coefficient | -- | 0.97 | 0.95 | 0.87 |





**Table 3. Statistical metrics of monthly mean precipitation deseasonalized anomalies among forcing data sets and CN05.1 during 2008-2014.**

| Metrics | CMFD-CN05.1 | CLDAS-CN05.1 | GLDAS-CN05.1 |
|---|---|---|---|
| RMSE(mm yr$^{-1}$) | 3.51 | 3.55 | 2.15 |
| Relative bias | -1.74 | 2.86 | 5.25 |
| Correlation coefficient | 0.88 | 0.89 | 0.95 |





**Table 4. Spatial statistical metrics of annual mean shortwave radiation and bias from 2008 to 2014 (CMFD was used as reference data when average difference, RMSE, relative bias and correlation coefficient were calculated).**

| Metrics | CMFD | CLDAS | GLDAS |
|---|---|---|---|
| Mean(W m$^{-2}$) | 178.60 | 202.26 | 203.13 |
| SD(W m$^{-2}$) | 31.13 | 28.82 | 20.99 |
| CV | 0.17 | 0.14 | 0.10 |
| Average difference (W m$^{-2}$) | -- | 23.68 | 24.55 |
| RMSE(W m$^{-2}$) | -- | 27.58 | 28.60 |
| Relative bias | -- | 0.14 | 0.15 |
| Correlation coefficient | -- | 0.89 | 0.92 |



**Table 5. Statistical metrics of monthly mean shortwave radiation deseasonalized anomalies among forcing data sets during 2008-2014.**

| Metrics | CLDAS-CMFD | GLDAS-CMFD |
|---|---|---|
| RMSE(W m$^{-2}$) | 5.14 | 5.79 |
| Relative bias | 1.14 | 1.66 |
| Correlation coefficient | 0.50 | 0.62 |





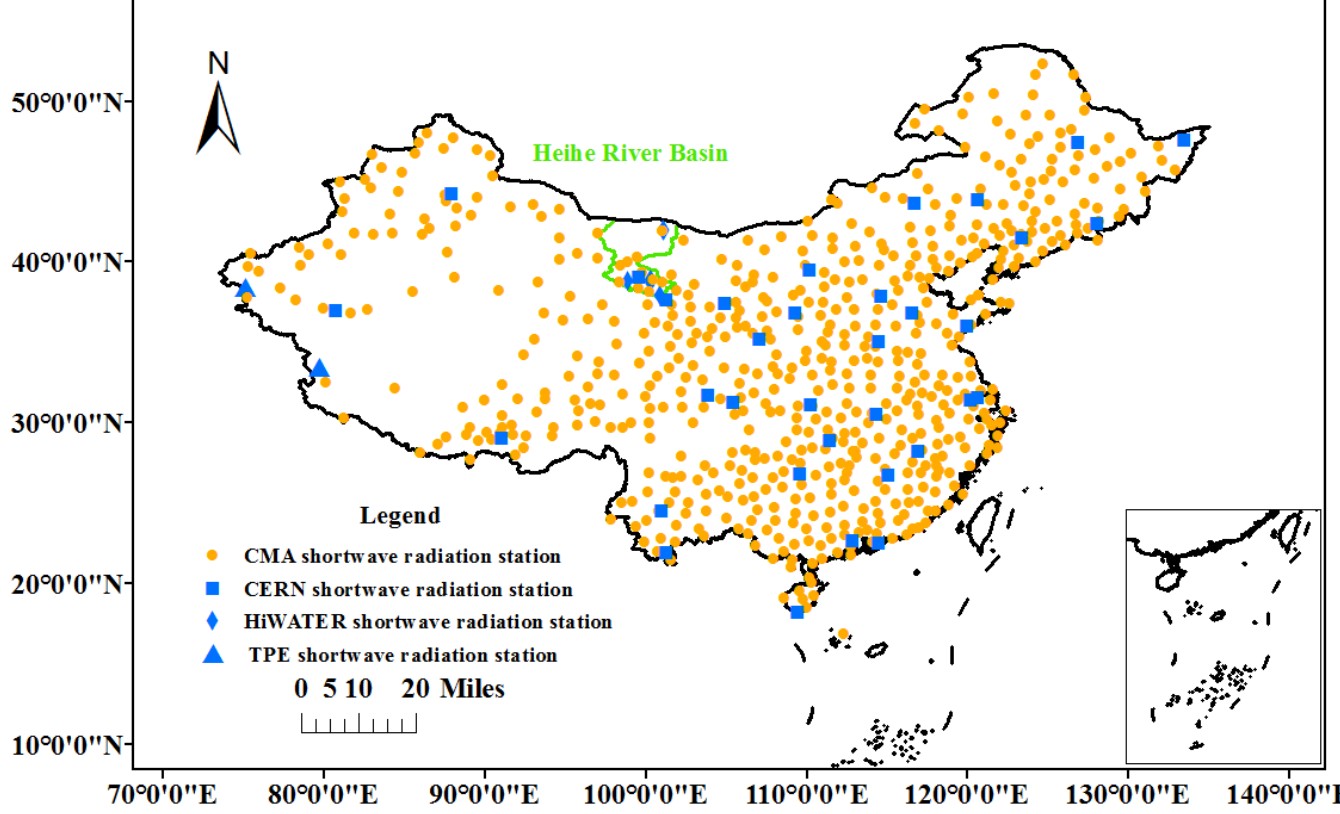

**Figure 1: Locations of stations over mainland China investigated in this study.**





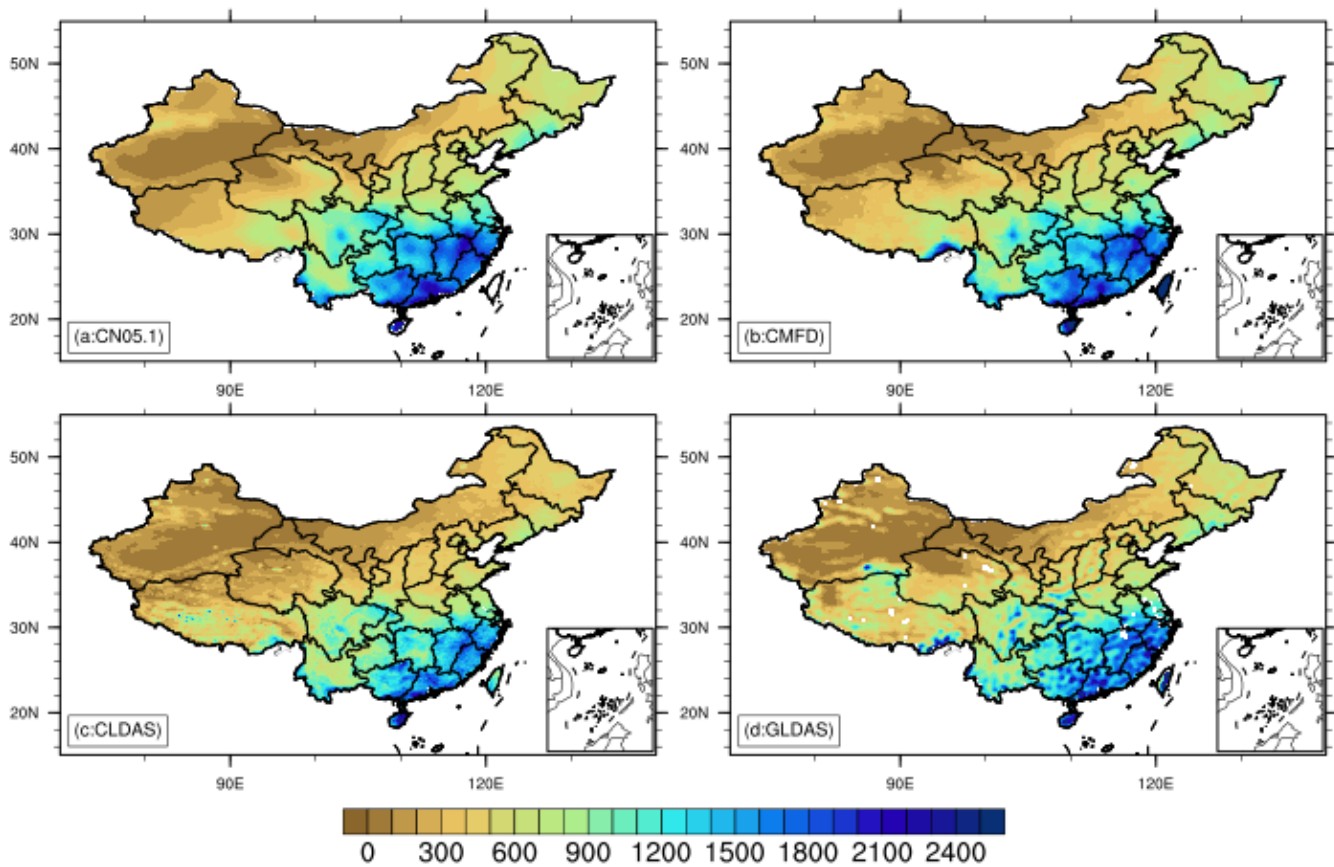

**Figure 2: Spatial distribution of annual mean precipitation (over 2008-2014, unit: mm yr-1).**





**Figure 3: Annual mean precipitation biases and corresponding histogram from CMFD (a and c), CLDAS (b and e) and GLDAS (c and f).**





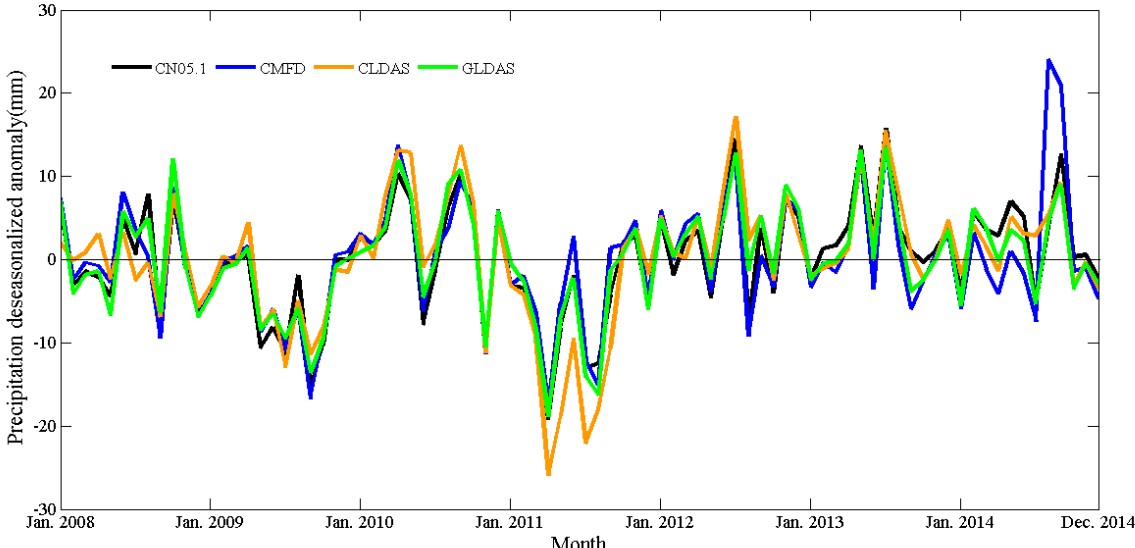

**Figure 4: Time series of monthly mean precipitation deseasonalized anomalies from CN05.1 (black), CMFD (blue), CLDAS (orange) and GLDAS (green).**





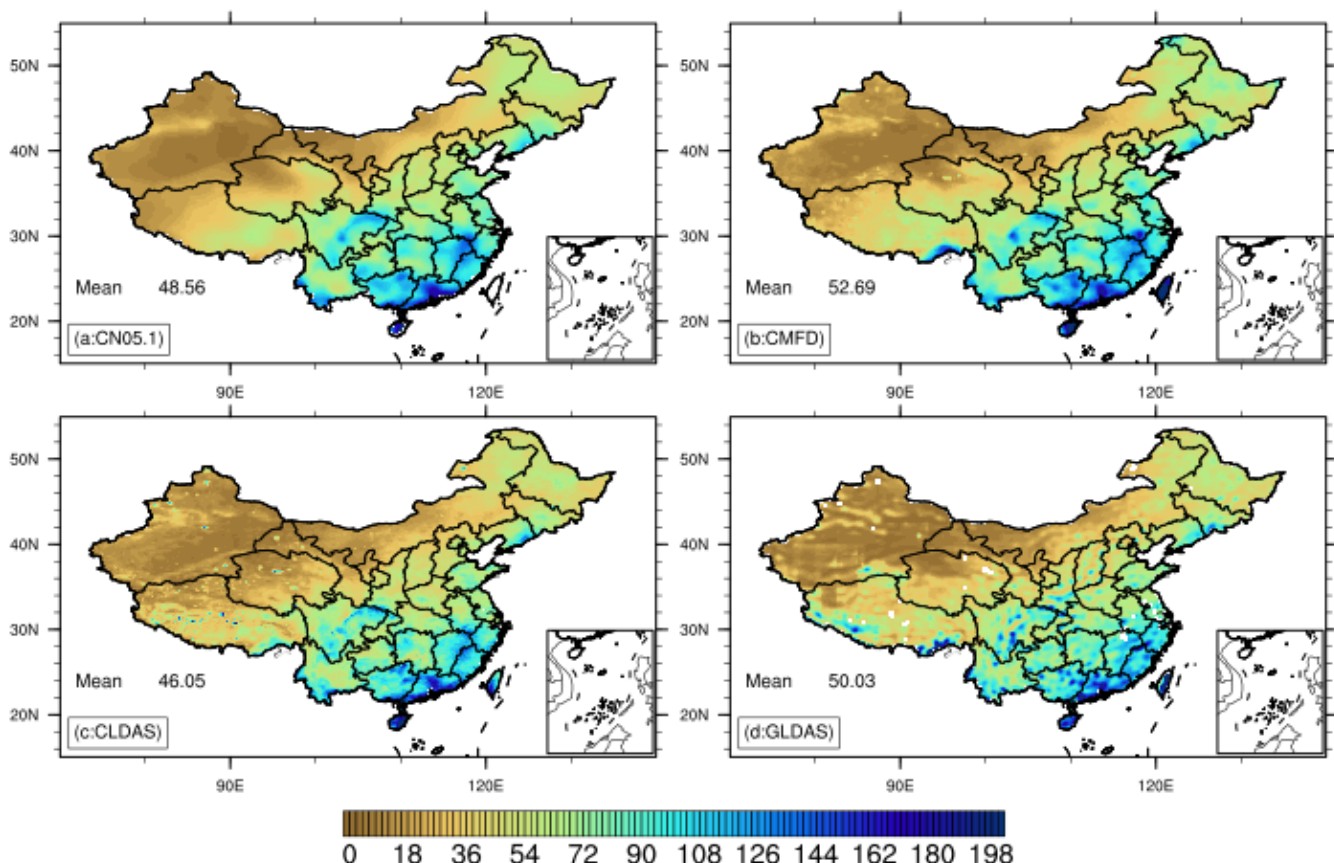

**Figure 5: Distribution of TSD of precipitation (over 2008-2014, unit: mm yr-1).**




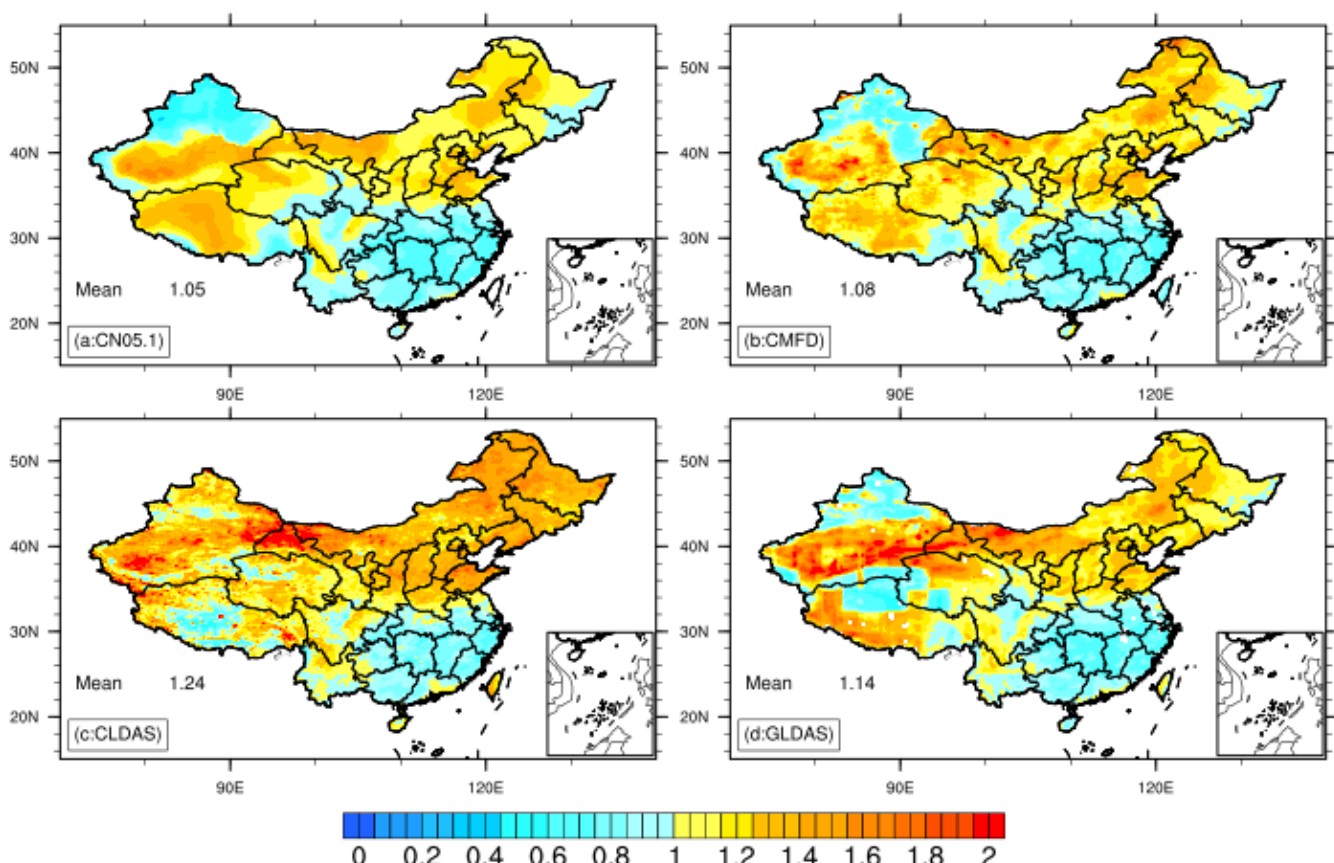

**Figure 6: Distribution of TCV of precipitation from 2008 to 2014.**






**Figure 7:** **The relationship between shortwave radiation from forcing data sets and observation data.**





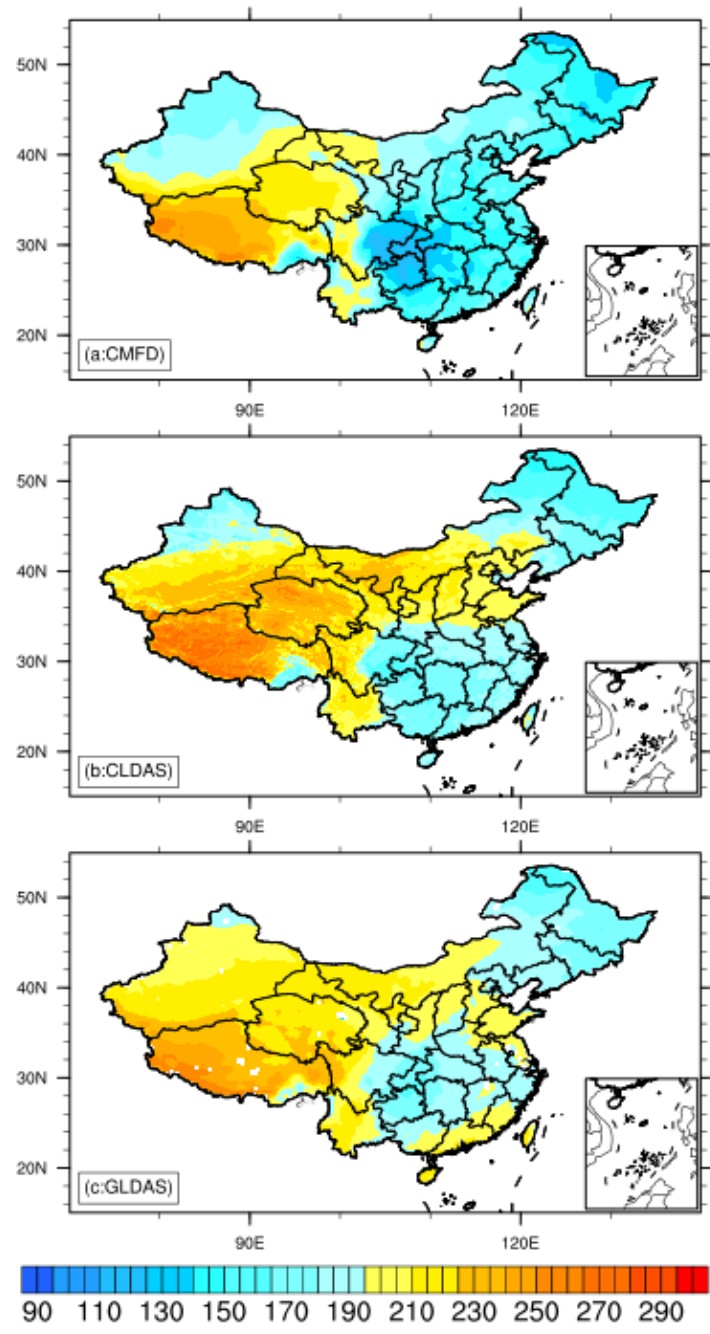

**Figure 8: Spatial distribution of annual mean shortwave radiation of three forcing data set (over 2008-2014, unit: W m-2).**





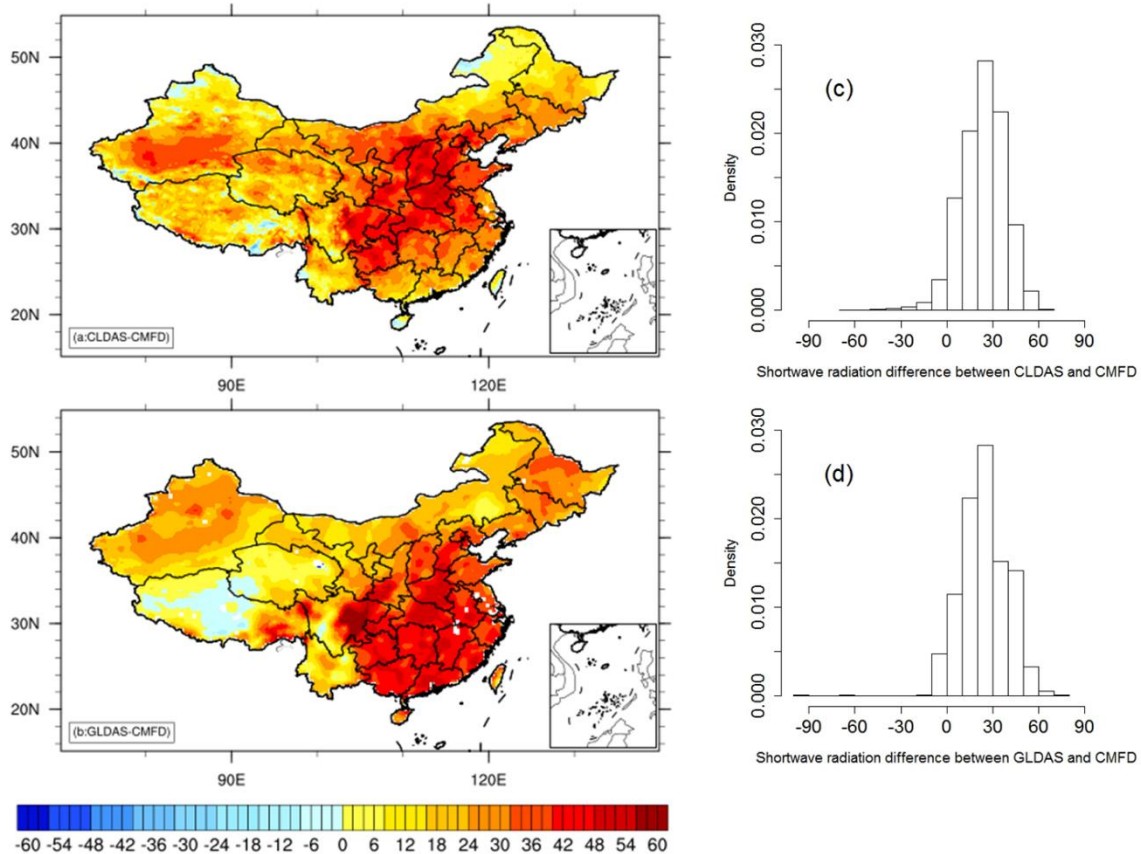

**Figure 9: Shortwave radiation difference among three forcing data sets and corresponding histogram from CMFD and CLDAS (a and c), GLDAS and CLDAS (b and e) and GLDAS and CMFD (c and f).**




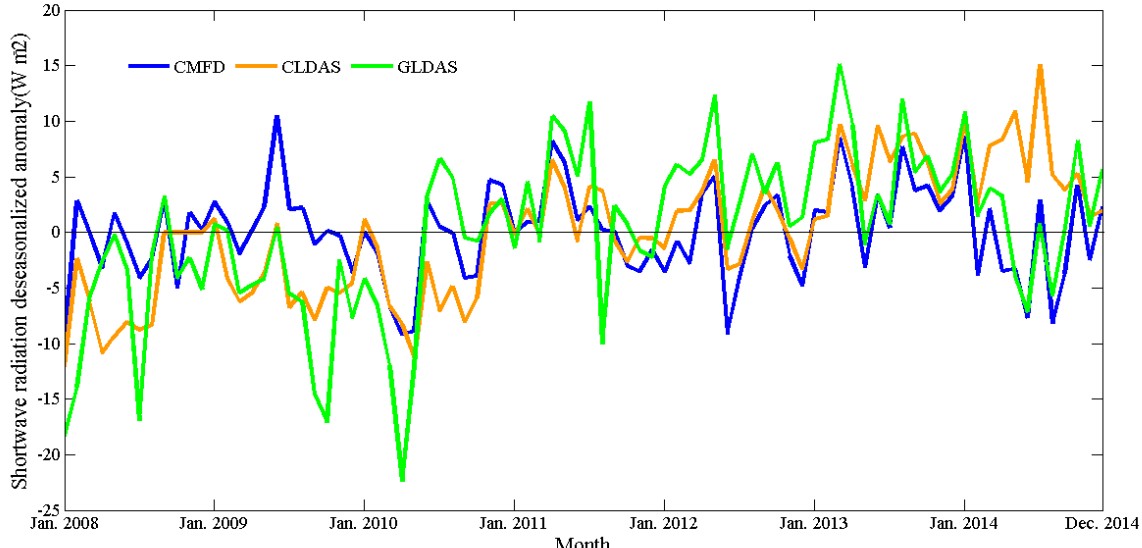

**Figure 10: Time series of monthly mean shortwave radiation deseasonalized anomalies from CMFD (blue), CLDAS (orange) and GLDAS (green) (unit: W m$^{-2}$).**



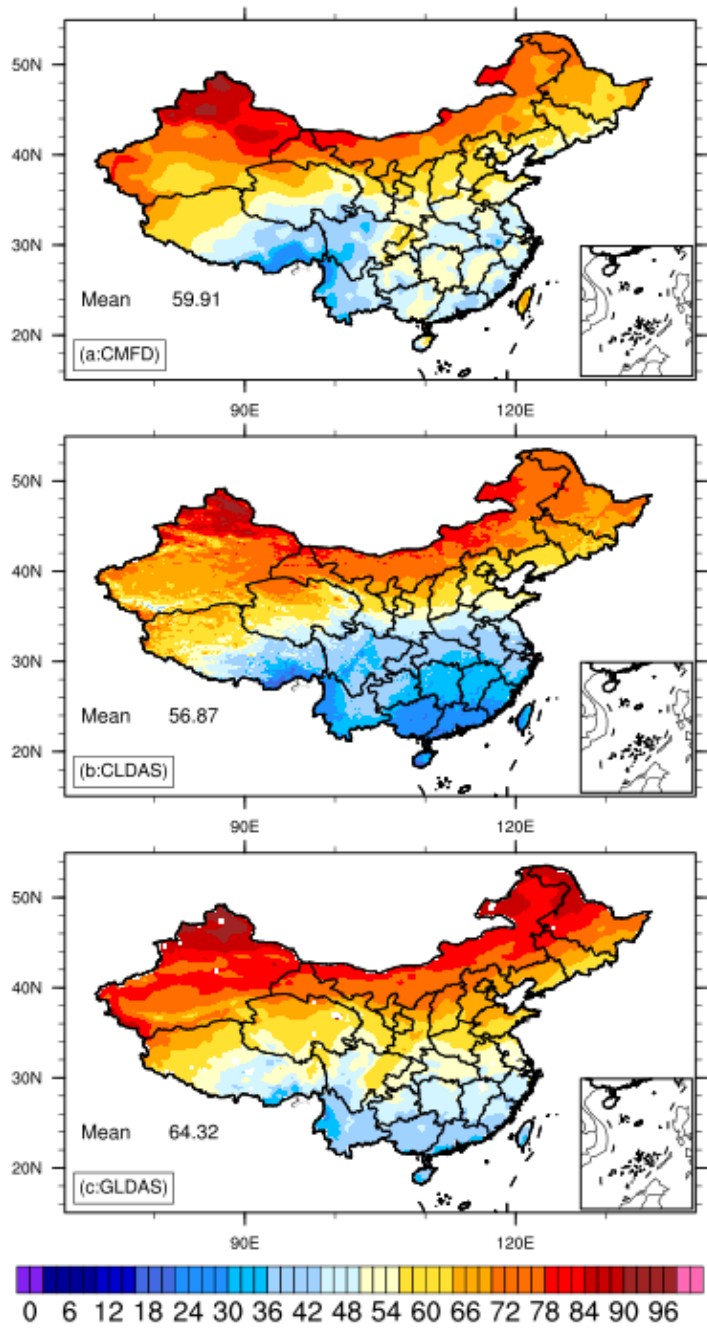

**Figure 11: Distribution of TSD (over 2008-2014, unit: W m-2) of shortwave radiation from forcing data sets.**





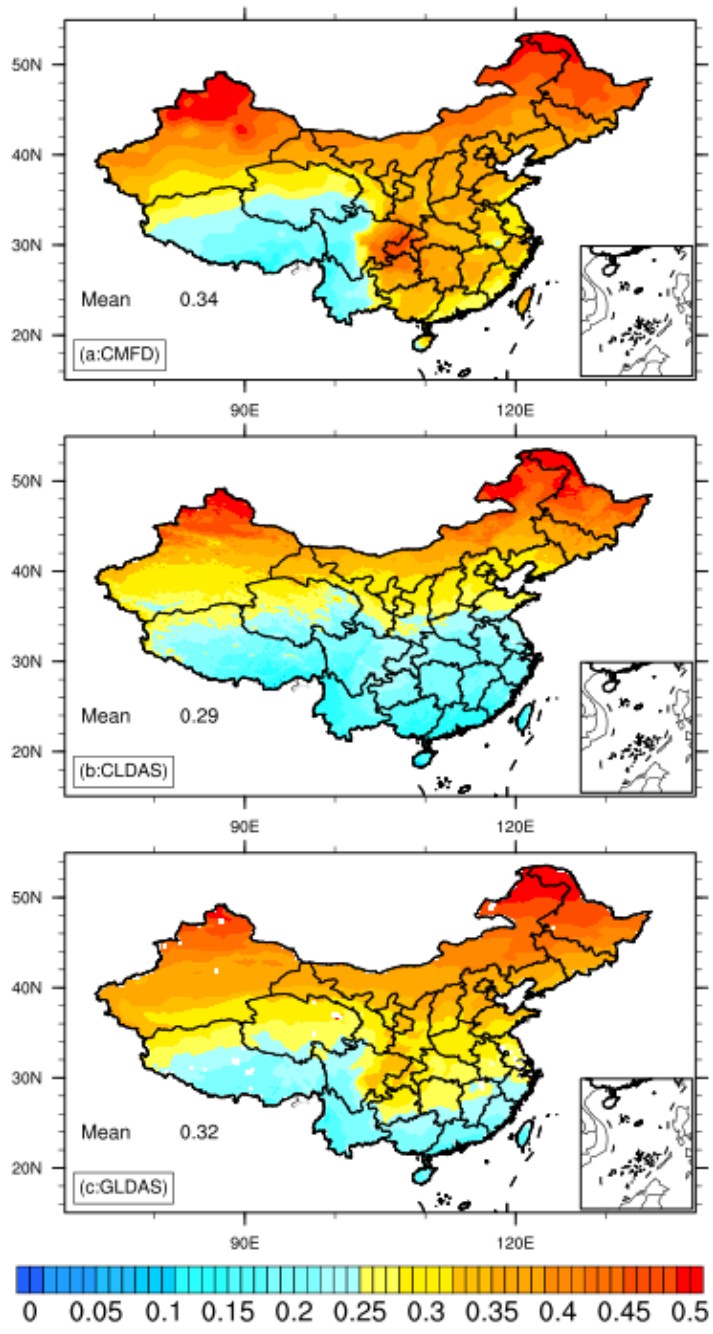

**Figure 12: Distribution of TCV of shortwave radiation from forcing data sets.**