# Peer review of "Evaluation of Multiple Forcing Data Sets for Precipitation and Shortwave Radiation over Mainland China"

_Hydrology and Earth System Sciences, 2017_

## Referee Comment (RC1) · Anonymous Referee #1 · 19 Jul 2017

This manuscript evaluated precipitation and shortwave radiation for three datasets: GLDAS, CLDAS and CMFD, where the latter two were created by Chinese scientists. My main concerns are the selection of the "truth" for evaluation, and the superficial analysis. The manuscript is not to create a new dataset, so simple comparison does not guarantee a publication in HESS. There are several comments below.

Major comments: 1. While the evaluation for radiation has some novelty given that the authors use several observation networks (CERN, HiWater and TPE) that are independent from the CMA data, the evaluation for precipitation is less credible. As far as I know, CLDAS used over 30,000 ground observation merged with CMORPH for precipitation, while the reference data used in this study (i.e., CN05.1) was only interpolated from 2000 stations. So, which one do you believe that is more close to the reality? I guess most people would buy CLDAS. In addition, CMFD was also basically interpolated from 740 stations, so it is not surprise that CMFD might be closer to CN05.1 than the CLDAS. So the fundamental issue is: which one is the truth for precipitation?

2. The second major issue is the resampling. As compared with GLDAS, the advantage of CLDAS and CMFD is their high resolution. So the evaluation should be conducted across scales, i.e., they should be verified against station observations besides at 0.25-degree resolution. Even for the gridded comparison conducted in the manuscript, the Nearest Neighbor resampling method is not suitable. The Nearest Neighbor is useful for comparing gridded data with station data, while for the comparison between two gridded datasets, bilinear interpolation or the inverse quadratic distance weighting method would be more appropriate.

3. The third major issue is the superficial analysis. An arbitrary conclusion that "In summary, precipitation estimates of CMFD and GLDAS are more credible and CMFD outperforms CLDAS and GLDAS in shortwave radiation estimation over mainland China" is definitely not enough for a HESS paper. The manuscript needs to answer why one dataset is better than another to provide more insights for the data production community. Alternatively, it should provide some in-depth comparison, for example, long-term trends, inter-annnual variability and extremes. The authors had a good start to show the temporal variations, but the new findings, if there are, should be concluded in the abstract.

Minor comments: 1. As the paper mainly focus on dataset evaluation, detailed introduction for different datasets should be provided to help distinguish them. For instance, CMFD uses GLDAS precipitation to replace TRMM 3B42 north to the 40°N which makes CMFD has the same background with GLDAS in that region. And, GLDAS also uses TRMM 3B42. Will this similarity influence the result?

2. P3L29, CLDAS-v2.0 covers the area of 60-140E and 0-65N. So I believe the manuscript used CLDAS-v1.0. Anyway, a more detailed description of CLDAS is needed.

3. P5L13, Xin et al. 2013 should be Li et al. 2013.

4. In terms of spatial comparison, are there any seasonal differences besides the annual precipitation? How about some daily statistics (e.g., rainfall frequency, intensity, dry spells)?

5. The radiation results are more convincing, and the different sources for three datasets make the comparison more meaningful. But some discussions can be given on why CMFD outperforms.

---

## Author Comment (AC1) · 20 Jul 2017

Dear Referee #1

Many thanks for your valuable suggestions and comments! We believe they will help to improve the quality of our manuscript and make it reach the criterion of HESS. In order to reply to your comments, we will use some new independent rainfall data, resample data, and make deep analysis about new results. Since it may take several weeks to get new results, here we give a quick reply to list what we will do.

1. About the precipitation data comparison. As referee pointed out that CLDAS,

[Figure]

CN05.1, and CMFD use CMA rain gauge data, and it is difficult to answer which one is the truth for precipitation. In order to answer this question, we will try to find an independent observation dataset of precipitation. 2. About resampling issue, we will follow referee's suggestion and resample gridded data with bilinear interpolation method and then conduct analysis again. 3. We will also follow referee's valuable suggestion about a comparison in temporal dimension. Moreover, we have invited Dr. Jie He, the developer of CMFD, as a co-author and cooperate together to analyze why CMFD performs better than others. At the same time, we will give more details about CLDAS and other data sets, which will also help to understand their discrepancy. Combining these together, we will have a deep analysis about the performance of new results. 4. We will also revise the minor problems pointed by referee #1.
* * *

---

## Referee Comment (RC2) · Anonymous Referee #2 · 4 Aug 2017

GLDAS, CLDAS and CMFD are the main forcing data over mainland China developed by different agencies. They can provide precipitation and shortwave radiation, which play important roles in climatic, hydrological and biogeochemical cycles. The correct evaluation and comparison between these dataset can help users to select the forcing data over mainland China. However, we can't see an objective evaluation among the three dataset because of the two concerns:

(1) The data used to evaluate the GLDAS, CLDAS and CMFD is not independent at all. This may lead to some mistake results. Actually, if you want to do an objective evaluation, independent data source is absolutely essential;

(2) The authors are not familiar with the techniques used to develop the GLDAS, CLDAS and CMFD. It can be seed from the wrong references cited by the authors. The author should correct these mistake in the future;

(3) The title "Evaluation and Comparison. . . . . . .". Some times, the Evaluation is included the Comparison;

(4) It is hard to determine which dataset is better than others if only judging from some traditional statistical metrics.

---

## Author Comment (AC2) · 23 Aug 2017

Dear Referee #2

We really appreciate your valuable comments which can help us revise the paper. Here we give a quick reply to explain what we will do in accordance with your comments. 1. About the data used to evaluate forcing data sets, we will follow the editor's advice and use the rain gauge data from Hydrological Bureau of Ministry of Water Resources which is an independent observation dataset of precipitation to evaluate forcing data sets. 2. About wrong references, we will check the references we cited carefully and correct these mistakes. 3. In order to refine the title, we can change it as "Evaluation

of Multiple Forcing Data Sets for Precipitation and Shortwave Radiation over Mainland China". 4. As for the statistical metrics, a Taylor diagram will be used to further describe the degree of correspondence between forcing data and observation data, and we will modify the statement in our manuscript to make it more objective.

Thanks again!

---

## Author Response (AR1)

Evaluation of multiple forcing data sets for precipitation and shortwave radiation over mainland China

by Fan Yang, Hui Lu, Kun Yang, Jie He, Wei Wang, Chengwei Li, Menglei Han, Yishan Li

We would like to thank the editor for your comments and suggestions, which improve the quality of our manuscript substantially. In this response, we first summarize major revisions, and then present the point-by-point response to comments of the individual referees.

**1 Major revisions**

1. According to the referees, independent data is necessary to do an objective evaluation. By following their comments and editor's suggestion, we obtained an independent precipitation data set observed by a rain gauge network operated by the Ministry of Water Resources (MWR), and then MWR data was used to evaluate the precipitation data from CN05.1, CMFD, CLDAS and GLDAS. The results of this independent evaluation were shown in the section4.3.

2. The CN05.1 precipitation data was treated as a forcing data set too. When we compared CN05.1 against MWR precipitation data, it was found that the CN05.1 differs obviously from the independent precipitation observations. Therefore, it is no longer used as reference data, but treaded as a forcing data generated by interpolating CMA station observations.

3. A new precipitation version of CMFD was used to replace to old one. Due to CMFD has updated their precipitation data recently, we made a new analysis of the new version. And in order to analyze deeply the performance of CMFD, we invited it producer, Dr. Jie He, as a co-author.

4. We added a part of discussion in the manuscript (section 6). According to the comments from referee#1, superficial analysis is not enough. As a result, we added some discussions on why the forcing data sets have such performance.

5. We removed a metric named TSD which represent the temporal standard deviations, because it is highly correlated with the temporal coefficient variation (TCV). In order to avoid redundancy, we only remained TCV in current version.

**2 Point-by Point response**

**2.1 Response to editor**

**Comment:**

1) The submission evaluated three datasets including GLDAS, CLDAS, CMFD over China. Such evaluations would be useful for the users to select proper dataset if deep insights can be obtained. Although a lot of work has been done, there are several major concerns raised in the reviewing process, of which the most serious one is the independent dataset from the three dataset. As far as I know, Hydrological Bureau of Ministry of Water Resources maintains a high density of rain gauge network. I suggest authors to use this independent dataset for the evaluation and submit a substantially revised manuscript.

**Response:**

Thanks for your advice to use the independent precipitation observation data provided by the Ministry of Water Resources (MWR). We obtained MWR precipitation data in 2014 observed by a rain gauge network located in the middle and lower reaches of the Yangtze River, where four forcing data show obvious differences. We added an introduction of the rain gauge network in Section 2.1.1, and also included the basic information of MWR data into Table 1. Moreover, we illustrated the location of this MWR network in Figure 1.

With this independent precipitation data set, CN05.1, CMFD, CLDAS and GLDAS were evaluated at annual and monthly scale. The evaluation method is the same as that used for shortwave radiation. CMFD and CLDAS were resampled to 0.25°×0.25° by the bilinear resampling method and the pixel-point method is applied when they are compared against MWR gauge observations.

[revised manuscript text omitted]

**2.2 Response to referee#1**

**1) Comment:**

While the evaluation for radiation has some novelty given that the authors use several observation networks (CERN, HiWater and TPE) that are independent from the CMA data, the evaluation for precipitation is less credible. As far as I know, CLDAS used over 30,000 ground observation merged with CMORPH for precipitation, while the reference data used in this study (i.e., CN05.1) was only interpolated from 2000 stations. So, which one do you believe that is more close to the reality? I guess most people would buy CLDAS. In addition, CMFD was also basically interpolated from 740 stations, so it is not surprise that CMFD might be closer to CN05.1 than the CLDAS. So the fundamental issue is: which one is the truth for precipitation?

**Response:**

We are very appreciate for your valuable comments about the reference data of precipitation. Accordingly, we have obtained an independent precipitation data set observed by a rain gauge network operated by the Ministry of Water Resource (MWR) of China. This MWR rainfall observations were used to evaluate the four forcing data sets. We added an introduction of the MWR rain gauge network in Section 2.1.1, and also included the basic information of MWR data into Table 1. Moreover, we illustrated the location of this MWR network in Figure 1. The evaluation results do show that CLDAS which merged more station data performs better than the other forcing data sets. For more details, please check section 4.3 and also the response to editor

**2) Comment:**

The second major issue is the resampling. As compared with GLDAS, the advantage of CLDAS and CMFD is their high resolution. So the evaluation should be conducted across scales, i.e., they should be verified against station observations besides at 0.25 degree resolution. Even for the gridded comparison conducted in the manuscript, the Nearest Neighbor resampling method is not suitable. The Nearest Neighbor is useful for comparing gridded data with station data, while for the comparison between two gridded datasets, bilinear interpolation or the inverse quadratic distance weighting method would be more appropriate.

**Response:**

Thanks for your advice! As you suggested, we adopted a bilinear interpolation method to rescale CLDAS and CMFD to the resolution of GLDAS, i.e., 0.25 degree, and then re-do the comparison and evaluation of forcing data sets.

[revised manuscript text omitted]

**3) Comment:**

The third major issue is the superficial analysis. An arbitrary conclusion that "In summary, precipitation estimates of CMFD and GLDAS are more credible and CMFD outperforms CLDAS and GLDAS in shortwave radiation estimation over mainland China" is definitely not enough for a HESS paper. The manuscript needs to answer why one dataset is better than another to provide more insights for the data production community. Alternatively, it should provide some in-depth comparison, for example, long-term trends, inter-annual variability and extremes. The authors had a good start to show the temporal variations, but the new findings, if there are, should be concluded in the abstract.

**Response:**

Thanks for your comment which helps us to improve the manuscript dramatically. We added a part of discussion and analysis the reason why the forcing data sets have such performance. In order to analyze deeply the performance of forcing data, we invited Dr. Jie He, main producer of CMFD, as a co-author to contribute his experiences and understandings in the revision.

"Based on the preceding analysis, we can see that though these forcing data sets have some common characteristics and can reflect the features of precipitation and shortwave radiation over mainland China, they have many difference due to different resolution, the various data they merged and the diverse algorithm they used. For precipitation, the spatial distribution of forcing data sets were compared and their quality in the middle and lower reaches of the Yangtze River were evaluated. CLDAS performs better both in annual and monthly scale, this is not surprise because CLDAS merges data at more than 30000 stations which can improve the data quality greatly. The precipitation of CMFD perform well at annual mean but not so at monthly scale, and its heavily decrease of precipitation after August 2014 is abnormal. As far as we concern, the CMFD used less precipitation station data than CN05.1 and CLDAS, which influences its quality in 2014. GLDAS as a global data, the precision in mainland China is limited due to the observation data of China merged in it may be not enough. Though both CMFD and GLDAS merged remote sensing data, they are not similar because the station data they used are different. As for CN05.1 which was made by purely station data and mathematical method, it is reasonable that it performs worse than other forcing data sets in station-sparse regions.

Comparing the forcing data sets and observation data, it was found that the shortwave radiation of CMFD perform better than the other two. The reason is that there are only about 100 radiation stations that were sparsely deployed in China since 1961 and the radiation observation data may be unusable because it often include erroneous values and missing data (Shi et al. 2008), therefore, the radiation observation data merged in CLDAS and GLDAS are limited. However, for CMFD, it merged the 50-year

data set of daily surface solar radiation at 716 CMA stations which was aforementioned in section 2.1.2. Though this data set is estimated by model, it is widely validated and its performance is pretty well. As a result, shortwave radiation of CMFD is closer to observation data".

Also, we modified the abstract which consists the results of temporal variations. "The results demonstrate that all the four forcing data sets can capture the spatial distribution characteristics of precipitation over mainland China while the annual mean precipitation of CLDAS is smaller than others in most area. The time series of precipitation anomaly from the forcing data sets also match well with each other expect CMFD after August 2014. All forcing data sets show the temporal variations in dry region are greater than wet region. Compared with the independent precipitation observation data provided by the Ministry of Water Resources (MWR) in the middle and lower reaches of the Yangtze River, CLDAS performs best and the annual mean precipitation of CMFD also match well with the MWR station data. However, GLDAS shows a large dispersion degree and CN05.1 obviously overestimates the precipitation with the highest bias in eight months. As for shortwave radiation, CMFD is consistent with observation data, while CLDAS and GLDAS heavily overestimate shortwave radiation when compared against station data. Spatially, the three forcing data sets have some common distribution features. Compared with CMFD, CLDAS and GLDAS have higher radiation values in most areas of mainland China. However, the metrics we calculated indicate that CLDAS performs better than GLDAS. For temporal variations, CLDAS is closer to CMFD than GLDAS, while their amplitude of anomalies are diverse. Also, the temporal variation difference of shortwave radiation from the three forcing data sets mainly exists the south of 34°N."

**4) Comment:**

As the paper mainly focus on dataset evaluation, detailed introduction for different datasets should be provided to help distinguish them. For instance, CMFD uses GLDAS precipitation to replace TRMM 3B42 north to the 40°N which makes CMFD has the same background with GLDAS in that region. And, GLDAS also uses TRMM 3B42. Will this similarity influence the result?

**Response:**

Many thanks for your comment and we revised the description of data sets used in this manuscript in section 2. "The CMFD forcing data set was developed by the Institute of Tibetan Plateau Research, Chinese Academy of Sciences (He and Yang, 2011). This product covers the region of 70.0°E -140.0°E and 15.0°N -55.0°N, and includes precipitation, downward   shortwave radiation, downward longwave radiation, 2-meter air temperature, specific humidity, wind speed and surface pressure. Tropical Rainfall Measuring Mission (TRMM) 3B42 precipitation data is used as the background field of precipitation data.

However, TRMM has no valid data in the north of 40°N in most of the time. Therefore, GLDAS is used in these regions to solve this problem. Gauge observation data from 740 stations of CMA are used to correct systematic deviations in background data. The Global Energy and Water cycle Experiment - Surface Radiation Budget (GEWEX-SRB) radiation data provide the background field for the shortwave radiation data of CMFD. Notably, GLDAS also used to replace GEWEX-SRB in its unavailable time and region. Shortwave radiation data estimated with CMA station data which has been mentioned in 2.1.2 is also used. Other basic information of the data is listed in Table 1.The 0.25°×0.25° monthly GLDAS-1 forcing data from the NOAH model (it is abbreviated as GLDAS in this paper) is provided by the US National Aeronautics and Space Administration (NASA). From 2001 to the present, this version makes use of National Oceanic and Atmospheric Administration (NOAA) Climate Prediction Center Merged Analysis of Precipitation (CMAP) fields, which merged satellite data (IR and microwave) and gauge data. CMAP fields are spatially and temporally disaggregated by Global Data Assimilation System (GDAS) modeled precipitation fields. A procedure and cloud and snow products from the Air Force Weather Agency (AFWA) Agricultural Meteorology modeling system (AGRMET) are used to calculate downward shortwave radiation fluxes (Rui and Beaudoing, 2017; Rodell et al. 2004)." Though both CMFD and GLDAS use TRMM 3B42 when they were product, the station data they used are not the same and their precipitation performance are different.

**5) Comment:**

P3L29, CLDAS-v2.0 covers the area of 60-140E and 0-65N. So I believe the manuscript used CLDAS-v1.0. Anyway, a more detailed description of CLDAS is needed.

**Response:**

We feel very sorry that we gave a wrong information about CLDAS in the last manuscript. We confirmed that the version we used in the manuscript is CLDAS-v2.0 whose spatial coverage is 60°E - 160°E, 0°N -65°N (L23, P5). In addition, we gave a more detailed description of CLDAS as you advised. "CLDAS-V2.0 was developed by CMA (Shi et al. 2014) and its spatial coverage is 60°E -160°E, 0°N - 65°N. This is hourly gridded data with a spatial resolution of 0.0625°×0.0625°. CLDAS includes land surface forcing data, such as precipitation, shortwave radiation, temperature, specific humidity, wind speed and surface pressure, as well as soil status variables. It is a relatively new product, with current temporal coverage from 2008 to 2017. Precipitation is combined and interpolated from two products, one is the Climate Prediction Center Morphing Technique (CMORPH) product and the other is an hourly merged precipitation product (V1.0) made by CMA which based on the observation data from automatic weather stations in China and CMORPH products through probability density function (PDF) and optimal

interpolation (OI) merging algorithm (Shen et al. 2014). Shortwave radiation is retrieved from the FY-2C/E series of geostationary meteorological satellites. The Discrete Ordinates Radiative Transfer Program for a Multi-Layered Plane-Parallel Medium (DISORT) method is used in the retrievals for radiation transfer calculations (Shi et al. 2011)."

**6) Comment:**

P5L13, Xin et al. 2013 should be Li et al. 2013.

**Response:**

Thanks for your comment, we corrected the mistake in L26, P4.

**7) Comment:**

In terms of spatial comparison, are there any seasonal differences besides the annual precipitation? How about some daily statistics (e.g., rainfall frequency, intensity, dry spells)?

**Response:**

We fell appreciate for your advice! However, we only have the independent precipitation observation data of the year 2014. Generally, winter in China covers from December to February in the following year, which mean that the CMA observation data in winter is unavailable. Besides, the CN05.1 and GLDAS used in this study is monthly scale data, some daily statistics cannot be provided. In order to make up these problem, we added the evaluation in monthly scale and the results we shown in Table 3 and section 4.3. "The evaluation results of monthly scale are listed in Table 3. According to the metrics, CLDAS performs best in most of the months of 2014."

**Table 3. Statistical metrics of monthly precipitation in 2014 between forcing data set and MWR rain gauge observations.**

| Time | Bias | | | | RMSE | | | |
|---|---|---|---|---|---|---|---|---|
| | CN05.1 | CMFD | CLDAS | GLDAS | CN05.1 | CMFD | CLDAS | GLDAS |
| Jan. | 23.62 | 2.37 | -1.85 | -0.48 | 29.53 | 19.70 | 8.90 | 8.26 |
| Feb. | 48.04 | 31.53 | 4.74 | 20.43 | 59.13 | 59.87 | 26.19 | 37.15 |
| Mar. | 61.37 | 14.81 | -9.26 | 12.50 | 77.02 | 37.54 | 36.43 | 59.47 |
| Apr. | 62.83 | 17.93 | -8.47 | 12.39 | 79.06 | 47.63 | 35.86 | 67.15 |
| May | 66.23 | 24.72 | 13.32 | 39.51 | 93.05 | 66.51 | 52.63 | 105.35 |
| June | 40.11 | 23.59 | 0.49 | 19.48 | 74.53 | 67.77 | 50.06 | 87.73 |
| July | 18.68 | 7.38 | -0.24 | 1.04 | 67.86 | 78.26 | 59.45 | 92.06 |
| Aug. | 22.37 | -53.29 | 3.57 | 11.01 | 62.51 | 78.93 | 47.46 | 74.30 |
| Sept. | 4.58 | -21.18 | -8.57 | 1.74 | 36.81 | 45.66 | 32.35 | 44.09 |
| Oct. | 26.07 | -17.59 | 5.56 | 8.72 | 37.11 | 38.95 | 27.76 | 32.35 |
| Nov. | 26.73 | -21.04 | -2.91 | 14.36 | 37.71 | 34.76 | 25.95 | 40.79 |
| Dec. | 8.99 | -5.43 | -4.33 | -1.65 | 14.90 | 16.97 | 9.40 | 10.84 |
| Annual | 421.38 | 273.68 | 234.39 | 427.71 | 403.16 | 3.80 | -7.93 | 139.06 |

**8) Comment:**

The radiation results are more convincing, and the different sources for three datasets make the comparison more meaningful. But some discussions can be given on why CMFD outperforms.

**Response:**

We gave an explanation in section 6. "The reason is that there are only about 100 radiation stations that were sparsely deployed in China since 1961 and the radiation observation data may be unusable because it often include erroneous values and missing data (Shi et al. 2008), therefore, the radiation observation data merged in CLDAS and GLDAS are limited. However, for CMFD, it merged the 50-year data set of daily surface solar radiation at 716 CMA stations which was aforementioned in section 2.1.2. Though this data set is estimated by model, it is widely validated and its performance is pretty well. As a result, shortwave radiation of CMFD is closer to observation data."

**2.3 Response to referee#2**

**1) Comment:**

The data used to evaluate the GLDAS, CLDAS and CMFD is not independent at all. This may lead to some mistake results. Actually, if you want to do an objective evaluation, independent data source is absolutely essential.

**Response:**

Thanks for your valuable comment! By following your comments and editor's suggestion, we have obtained an independent precipitation data set observed by a rain gauge network operated by the Ministry of Water Resource (MWR) of China. We used this independent MWR precipitation observation data to evaluate forcing data sets and the new results was shown in section 4.3, "the annual mean precipitation data of CMFD and CLDAS are more consistent with the MWR observation. However, the performance of GLDAS and CN05.1 are not as good as others. Their RMSE are about twice bigger than that of CLDAS. Also, it is obvious that the dispersion degree of GLDAS is the biggest compared with other data sets which indicate that GLDAS changes greatly in spatial. From the pattern of Fig. 6 (a) and the high bias of CN05.1, we can conclude that the annual mean precipitation in Hubei, Hunan and Jiangxi province are heavily overestimated by CN05.1. The evaluation results of monthly precipitation are listed in Table 3. According to the metrics, CLDAS performs best in most of the months of 2014. Fig. 7 also confirms that CLDAS performs well because the orange points representing CLDAS are concentrated together, located in a region where the correlation coefficient is between 0.6 and 0.9, the standardized deviation is close to 1 and the unbiased RMSE is low. This reflects that the quality of monthly precipitation of CLDAS is stable and reliable. However, the performance of other data sets in the monthly scale varies greatly, especially for CMFD and CN05.1." We gave a detailed explanation in the response to editor.

**Table 3. Statistical metrics of monthly precipitation in 2014 between forcing data set and MWR rain gauge observations.**

| Time | Bias | | | | RMSE | | | |
|---|---|---|---|---|---|---|---|---|
| | CN05.1 | CMFD | CLDAS | GLDAS | CN05.1 | CMFD | CLDAS | GLDAS |
| Jan. | 23.62 | 2.37 | -1.85 | -0.48 | 29.53 | 19.70 | 8.90 | 8.26 |
| Feb. | 48.04 | 31.53 | 4.74 | 20.43 | 59.13 | 59.87 | 26.19 | 37.15 |
| Mar. | 61.37 | 14.81 | -9.26 | 12.50 | 77.02 | 37.54 | 36.43 | 59.47 |
| Apr. | 62.83 | 17.93 | -8.47 | 12.39 | 79.06 | 47.63 | 35.86 | 67.15 |
| May | 66.23 | 24.72 | 13.32 | 39.51 | 93.05 | 66.51 | 52.63 | 105.35 |
| June | 40.11 | 23.59 | 0.49 | 19.48 | 74.53 | 67.77 | 50.06 | 87.73 |
| July | 18.68 | 7.38 | -0.24 | 1.04 | 67.86 | 78.26 | 59.45 | 92.06 |
| Aug. | 22.37 | -53.29 | 3.57 | 11.01 | 62.51 | 78.93 | 47.46 | 74.30 |
| Sept. | 4.58 | -21.18 | -8.57 | 1.74 | 36.81 | 45.66 | 32.35 | 44.09 |
| Oct. | 26.07 | -17.59 | 5.56 | 8.72 | 37.11 | 38.95 | 27.76 | 32.35 |
| Nov. | 26.73 | -21.04 | -2.91 | 14.36 | 37.71 | 34.76 | 25.95 | 40.79 |
| Dec. | 8.99 | -5.43 | -4.33 | -1.65 | 14.90 | 16.97 | 9.40 | 10.84 |
| Annual | 421.38 | 273.68 | 234.39 | 427.71 | 403.16 | 3.80 | -7.93 | 139.06 |

[Figure]

**Figure 6: Comparison of the precipitation from (a) CN05.1, (b) CMFD, (c) CLDAS, and (d) GLDAS against MWR rain gauge observation. The color bar on the right indicates the number of MWR rain gauges in one 0.25°×0.25° grid.**

[Figure]

**Figure 7: Taylor diagram for the monthly/annual precipitation of CN05.1 (black), CMFD (blue), CLDAS (orange), and GLDAS (green).**

**2) Comment:**

The authors are not familiar with the techniques used to develop the GLDAS, CLDAS and CMFD. It can be seed from the wrong references cited by the authors. The author should correct these mistake in the future.

**Response:**

We need apologize for the wrong references in our manuscript. We modified the instruction of data sets we used and corrected the references in section 2. Moreover, in order to analyze the performance of CMFD deeply, we invited another author of CMFD (Kun Yang is one author of it), Dr. Jie He, as a co-author to discuss the techniques used in developing forcing data.

"The CMFD forcing data set was developed by the Institute of Tibetan Plateau Research, Chinese Academy of Sciences (He and Yang, 2011). This product covers the region of 70.0°E -140.0°E and 15.0°N -55.0°N, and includes precipitation, downward shortwave radiation, downward longwave radiation, 2-meter air temperature, specific humidity, wind speed and surface pressure. Tropical Rainfall

Measuring Mission (TRMM) 3B42 precipitation data is used as the background field of precipitation data. However, TRMM has no valid data in the north of 40°N in most of the time. Therefore, GLDAS is used in these regions to solve this problem. Gauge observation data from 740 stations of CMA are used to correct systematic deviations in background data. The Global Energy and Water cycle Experiment - Surface Radiation Budget (GEWEX-SRB) radiation data provide the background field for the shortwave radiation data of CMFD. Notably, GLDAS also used to replace GEWEX-SRB in its unavailable time and region. Shortwave radiation data estimated with CMA station data which has been mentioned in 2.1.2 is also used. Other basic information of the data is listed in Table 1.

CLDAS-V2.0 was developed by CMA (Shi et al. 2014) and its spatial coverage is 60°E -160°E, 0°N -65°N. This is hourly gridded data with a spatial resolution of 0.0625°×0.0625°. CLDAS includes land surface forcing data, such as precipitation, shortwave radiation, temperature, specific humidity, wind speed and surface pressure, as well as soil status variables. It is a relatively new product, with current temporal coverage from 2008 to 2017. Precipitation is combined and interpolated from two products, one is the Climate Prediction Center Morphing Technique (CMORPH) product and the other is an hourly merged precipitation product (V1.0) made by CMA which based on the observation data from automatic weather stations in China and CMORPH products through probability density function (PDF) and optimal interpolation (OI) merging algorithm (Shen et al. 2014). Shortwave radiation is retrieved from the FY-2C/E series of geostationary meteorological satellites. The Discrete Ordinates Radiative Transfer Program for a Multi-Layered Plane-Parallel Medium (DISORT) method is used in the retrievals for radiation transfer calculations (Shi et al. 2011).

The 0.25°×0.25° monthly GLDAS-1 forcing data from the NOAH model is provided by the US National Aeronautics and Space Administration (NASA). From 2001 to the present, this version makes use of National Oceanic and Atmospheric Administration (NOAA) Climate Prediction Center Merged Analysis of Precipitation (CMAP) fields, which merged satellite data (IR and microwave) and gauge data. CMAP fields are spatially and temporally disaggregated by Global Data Assimilation System (GDAS) modeled precipitation fields. A procedure and cloud and snow products from the Air Force Weather Agency (AFWA) Agricultural Meteorology modeling system (AGRMET) are used to calculate downward shortwave radiation fluxes (Rui and Beaudoing, 2017; Rodell et al. 2004.).”

**3) Comment:**

The title "Evaluation and Comparison……". Some times, the Evaluation is included the Comparison.

**Response:**

Thank you for the advice! We changed the title as "Evaluation of Multiple Forcing Data Sets for

Precipitation and Shortwave Radiation over Mainland China".

**4) Comment:**

It is hard to determine which dataset is better than others if only judging from some traditional statistical metrics.

**Response:**

Very appreciate! We added the Taylor diagram to further describe the degree of correspondence between forcing data and observation data. It shows the ratio of standardized deviations, correlation coefficient and unbiased RMSE between forcing data and observation data, and these statistics can quantify how closely the forcing data resembles the observation. As for precipitation, "Fig. 7 also confirms that CLDAS performs well because the orange points representing CLDAS are concentrated together, located in a region where the correlation coefficient is between 0.6 and 0.9, the standardized deviation is close to 1 and the unbiased RMSE is low. This reflects that the quality of monthly precipitation of CLDAS is stable and reliable. However, the performance of other data sets in the monthly scale varies greatly, especially for CMFD and CN05.1." As for shortwave radiation, compared with CMA station data, "Fig. 9 show that the unbiased RMSE between CMFD and CMA stations is the smallest and the correlation coefficient is the highest, the standardized deviation ratio is the closest to 1. These metrics of CLDAS perform better than GLDAS which indicate that CLDAS is more resemble to the observation than GLDAS. " Compared with CERN station data, "the metrics of CMFD shown in Fig. 9 perform the best which indicates that the estimation of CMFD for shortwave radiation is more precise than for CLDAS and GLDAS in these areas, and GLDAS is worse compared with CLDAS." Compared with the eight observation stations in the Heihe River basin and two observation stations in the Tibetan Plateau, "CLDAS has the smallest unbiased RMSE and the closest standardized deviation compared to the observation. The correlation coefficient between GLDAS and observation is the highest followed by CMFD and CLDAS, while the RMSE and relative bias of CLDAS and GLDAS are about 2 and 10 times that of CMFD, respectively."

[Figure]

**Figure 7: Taylor diagram for the monthly/annual precipitation of CN05.1 (black), CMFD (blue), CLDAS (orange), and GLDAS (green).**

[Figure]

Figure 9: Taylor diagram for the shortwave radiation of CMFD (blue), CLDAS (orange), and GLDAS (green).

**3 List of all relevant changes made in the manuscript**

1) Title: Delete "comparerion".

2) P1, L17-32: Modified the abstract as the referee#1 advised.

3) Section 2.1.1: Added an introduction of independent precipitation observation data.

4) Section 2.2: Added some information about the forcing data sets and corrected the mistakes as the referee pointed.

5) Section 3 (P6, L20): Changed the resample method and the results were shown in P9, L26-31.

6) Section 3 (P6, L23-26): Added Taylor diagram and the results were shown in Fig. 7 and Fig. 9.

7) Section 4.1 and 4.2: Evaluated the updated precipitation of CMFD and regard CN05.1 as other forcing data sets instead of the reference data.

8) Section 4.3: The independent precipitation observation data were used to evaluate forcing data sets in annual and monthly scale.

9) Section 6: Added the part of discussion to analyze why the forcing data sets have such performance.

[revised manuscript text omitted]

---

## Author Response (AR2)

**Editor, Hydrology and Earth System Science**

Dear Dr. Tian,

Many thanks for your coordination! And we feel very appreciate to the referees for their valuable comments and suggestions. Based on the comment raised by reviews and you, we have completed the revision of the following paper:

Manuscript No.: hess-2017-321

Title: Evaluation of multiple forcing data sets for precipitation and shortwave radiation over mainland China

Authors: Fan Yang, Hui Lu, Kun Yang, Jie He, Wei Wang, Jonathon S.Wright, Chengwei Li, Menglei Han, Yishan Li

First of all, we have revisied our manuscript according to the comments of Referee#3 and you. The point-to-point replies are listed as follows:

(1) *Comment*: I suggest the authors to have the paper been checked by a native English speaker for any grammar errors.

*Response*: Done. We have invited a native English speaker to proof the language of the revised version.

(2) *Comment*: Page 2 Line 27, "Gao and Wang, 2013" should be "Guo and Wang, 2013". Please check and correct any other citation errors.

*Response*: Many thanks for pointed out this typo! We have correct the mistake in Page2 Line 27.

(3) *Comment*: Please consider switching the order of section 2.1 and section 2.2. Perhaps it's better call the "observation data" as "validation data set".

*Response:* We have followed the referee's advice and exchanged the order of section 2.1 and section 2.2, and change the expression of observation data in Page 5 Line 3.

(4) *Comment*: Please consider including an additional table to summarize these ground measurements for shortwave radiation, like CERN, HiWATER and TPE data sets.

*Response*: Great idea. We added a table about the validation data set of shortwave radiation in Page 19.

(5) *Comment*: Page 6 Lines 4-5, "The 0.25°×0.25° monthly GLDAS-1 forcing data from the NOAH model (abbreviated as GLDAS in this paper) is provided by the US National Aeronautics and Space Administration (NASA)", this is wrong, the GLDAS forcing was not produced by Noah model.

*Response*: This mistake has been corrected in Page 4 Lines 27-28, "The 0.25°×0.25° monthly GLDAS-1 forcing data (abbreviated as GLDAS in this paper) is provided by the US National Aeronautics and Space Administration (NASA)."

(6) *Comment*: Figures 1 and 2, please remove the 0'0" in the axes.

*Response*: We modified these two figures and the new figures are shown in Page 25-26.

Sencondly, we would like to ask you to help us to add two co-authors of this paper. During the period of paper discussion, in order to explore the details of four forcing data sets, we invited Dr. Jie He (email: hejie.1207@gmail.com; address: Institute of Tibetan Plateau Research, Chinese Academy of Sciences), one of the producers of CMFD, and Dr. Jonathon S.Wright (email: jswright@tsinghua.edu.cn; address:Department of Earth System Science, Tsinghua University), an expert in reanalysis data application, to finalize our manuscript. They have contributed greatly to

improve the quality of our manuscript, therefore, we would like to include them as co-authors in the final paper.

Following is the revised version of our manuscript, we hope you will find that it is much improved.

Sincerely yours,

Dr. Hui Lu

Department of Earth System Science, Tsinghua University

Email: luhui@tsinghua.edu.cn

[revised manuscript text omitted]